# Trade-offs in cotton pest management: Seed treatments suppress pests but reduce the abundance of natural enemies in the arthropod community

Melis Yalçin  *

Department of Plant Protection, Faculty of Agriculture, Aydın Adnan Menderes University, Aydın, Türkiye

* melisusluy@adu.edu.tr

## Abstract

Pesticide seed treatments are widely used to protect crops from early-season pests and diseases; however, their broader ecological effects on arthropod communities and non-target organisms remain insufficiently understood. A two-year field experiment was conducted in a cotton agroecosystem to evaluate the community-level effects of seed-applied clothianidin (CLO) and a fungicide mixture containing azoxystrobin, metalaxyl-m, and fludioxonil (AMF). Changes in arthropod population density and diversity were assessed across taxonomic and community levels. The dominant arthropod families recorded included Nabidae, Miridae, Anthocoridae, Asilidae, Chrysopidae, Cicadellidae, and Coccinellidae. Both seed treatments suppressed key sucking pest groups, particularly Cicadellidae and Miridae; however, a significant reduction in predator abundance was also observed in treated plots. These findings indicate a potential trade-off between effective pest suppression and the conservation of natural enemy populations within the arthropod community. Principal response curve (PRC) analysis further demonstrated treatment-related shifts in arthropod communities over time. The results highlight the importance of considering non-target ecological responses when evaluating the sustainability of prophylactic seed treatments in cotton agroecosystems and emphasize the need for long-term investigations into their impacts on multitrophic interactions.

## Introduction

Cotton (*Gossypium hirsutum* L.) is a significantly important cash crop grown in a hundred countries across the globe [1,2] and an indispensable natural textile fiber commodity, often called white gold, and the king of fiber [3]. It fulfills one-third of the natural fiber demand of the world population [4]. However, cotton production faces serious threats from diverse arthropod pests and diseases. The damage caused by soil-borne diseases, early-season sap-sucking insect pests, and other pests to cotton

**Data availability statement:** All relevant data are within the manuscript and its Supporting Information files.

**Funding:** The author(s) received no specific funding for this work.

**Competing interests:** The authors have declared that no competing interests exist.

crops require sustainable pest management strategies to mitigate economic and crop production losses. Seed coating is one of the most crucial management strategies to protect seeds and seedlings. This technique was first used in 1930 for cereal seeds to prevent damage from soil-borne pathogens and became widespread between the 1970s and 1980s on field crops such as soybean, corn, wheat, and cotton [5,6].

Clothianidin (CLO) is a soil-active neonicotinoid insecticide commonly used for soil and plant pests. CLO targets nicotinic acetylcholine receptors (nAChR) and promotes paralysis and hyperstimulation of pests [7,8]. It is used to control black cutworm *Agrotis ipsilon*, wireworms from Coleoptera: Elateridae and Coleoptera: Tenebrionidae, White grubs, Coleoptera: Scarabaeidae, Cotton Flea beetle Coleoptera: Chrysomelidae, Homoptera: Aphididae, and Thysanoptera: Thripidae. Azoxystrobin is a synthetic, biodegradable, translaminar strobilurin fungicide [9]. As a mode of action, it can inhibit mitochondrial respiration between cytochromes [10]. It is generally combined with metalaxyl-m and fludioxonil. AMF is generally used on cotton for *Rhizoctonia solani*, *Fusarium*, and *Phytium*. The chemical properties of these pesticides affect their uptake and translocation in the plant. The lipophilicity, volatility, and polarity of the pesticide components and the vegetation period of the products are crucial for calculating the risk assessment and determining the amount of pesticides that are transferred through different parts of the plant [11–13].

Natural enemies may be exposed to insecticides and fungicides directly or indirectly through consumption of herbivores, contact with treated plant surfaces, or feeding on nectar, extrafloral nectar, pollen, honeydew, and guttation liquid [11,14–17]. Adverse effects have been reported for several taxa, including Coccinellidae, Anthocoridae, Nabidae, Chrysopidae, spiders, Braconidae, and Syrphidae [5,11,17–20]. Such exposure can reduce natural enemy abundance and cause non-target effects, particularly during the flowering period when nectar and pollen attract beneficial arthropods and guttation fluid provides water resources [21,22]. For example, *Apis mellifera* exposed to clothianidin contaminated guttation fluid from maize plants exhibited high mortality, indicating that systemic insecticides can affect non-target beneficial insects through plant-derived exposure routes [23]. Clothianidin seed treatments may also indirectly affect natural enemies by reducing prey availability, sometimes leading to pest outbreaks such as increased *Tetranychus* sp. populations [24]. Although some benefits have been reported, studies demonstrate that seed treatments with clothianidin, imidacloprid, or thiamethoxam can disrupt arthropod community structure, alter diversity and evenness, and decrease beneficial predator abundance [14,25]. Still, the effects of seed-applied pesticides on arthropod communities under field conditions are not fully understood.

This study examined how seed-applied pesticides influence sucking pest populations, a major issue in cotton agroecosystems, and their natural enemies at the community level. The abundance and diversity of both pests and predators were evaluated. Additionally, clothianidin (CLO) and a fungicide mixture (AMF) were applied as seed treatments to assess their efficacy and compare arthropod population densities across different pesticide applications. The findings provide new insights into the effects of seed treatments on cotton sucking pests and their potential side effects on predator communities.

## Materials and methods

### Study site

The study was conducted at the Aydın Adnan Menderes University Faculty of Agriculture Research and Application Farm (37°45′32″N, 27°45′35″E), Aydın, Turkey. Lodos cotton cultivar (Özaltın Seed Company of Turkey) was planted in the last week of May in 2021 and 2022, with 70 cm inter-rows and 15 cm intra-row spacing. The treatments were arranged in 20 and 10 m by a randomized complete block design with three replications. Each plot column was separated to ensure distinct boundaries and simplify sample collection; 1-meter bare strips were used to separate the columns. CLO was applied as a seed treatment (Poncho 600 Basf) at a rate of 800 ml 100 kg$^{-1}$ seed, and AMF was applied as a seed treatment (Dynasty CST 125 FS Syngenta) at a rate of 250 ml 100 kg$^{-1}$ seed, and also untreated seed as a control was used at the exact locations for each year. During the experiment, no pesticides were applied. Weeds were controlled with cultural practices (mainly tillage and hand-hoeing) during the growing season.

### Experimental setup and arthropod collection methods

Foliar-sucking arthropods and their predators were sampled for twelve weeks between 18 June 2021 and 7 September 2021, and 17 June 2022 and 9 September 2022 until the cotton harvest time. The average physical conditions between seeding and harvest were found 0.02 mm and 1.1 mm of rainfall, 26.9 °C (max. 34.9 °C and min.17.8 °C), 26.3 °C (max. 33.5 °C and min.18.3 °C), and 53.33% R.H. (max. 89%, min. 33.6%), 66.44% R.H. (98.7%, min. 37.7%) (Meteorological Station, 2021–2022 Aydın, Turkey). Samples were collected 12 times during the cotton growing season because the most active time interval for pests and predators is from June to September. Arthropod abundance in the plant canopy was measured by sweep netting, where 50 sweeps were taken in a straight line through the center of each plot. Sweep net sampling was used to assess the abundance of mobile foliar-dwelling arthropods present in the plant canopy. Sweep netting was conducted at intervals of 100 meters from the field's edge to avoid potential edge effects on the experimental results. Pests and predators were identified under a stereomicroscope (Leica E24, 10x) and recorded at the family level. Data from samples within three replicates were averaged for analysis. The interactions between the plant's vegetative stage and pesticide treatment effects were observed for both years in 2021 and 2022 in the Aydın cotton province.

### Statistical analysis

Arthropod diversity in the samples was assessed by identifying arthropods at the family level, with both adults and immature stages grouped for each taxon. Due to taxonomic uncertainty at the family level, the analyses did not include insects belonging to Diptera, Hymenoptera, and Lepidoptera.

The effects of seed treatment with CLO and AMF on the number of arthropods were determined using a general linear model (GLM). The sum of mean values of pests and natural enemies were compared using one-way ANOVA in SPSS, followed by Tukey's HSD post hoc test.

The total number of taxonomic families (Taxa richness), number of individuals recorded (Total abundance), Shannon index, and evenness were calculated to quantify the diversity of the arthropod community in the treated and untreated plots for each week.

The Shannon-Wiener Diversity Index (H) is the most used diversity index in ecological experiments [26,27].

$$H = -\sum_{i=1}^{s} Pi \ln Pi$$

In the formula 'ln'is the natural log, 'pi' is the proportion of individuals in the sum of the species.'∑' is the sum of the results. 's' is the total number of species.

Shannon Index and Evenness data were subjected to one-way ANOVA and compared using the LSD (least significant difference) test at the 0.05 probability level.

The overall composition of arthropod communities is based on each taxonomic group's abundance relative to the total number of arthropods recorded. Principal Response Curve (PRC) analysis was employed to synthesize overall community-level responses of arthropods to seed treatments. Sample data was averaged for each plot, and a two-year data set was included in the same analysis. Experimental design and synchronized sampling based on cotton growth enabled valid comparisons across years. As key factors to explain potential changes for each community dataset, seed treatment and sampling time were used, and data were log(x) transformed to adjust the raw data to stabilize the effect of variance. Canonical coefficients were obtained in R (R Core Team, 2012) using the 'vegan' package [28,29] for each sampling date to track the arthropod community response to the seed treatment over time. These coefficients were plotted over time, revealing how the seed-treated community differed from the control group throughout the experiment. The arthropod group most affected by the seed treatment was computed according to taxon-specific weights. A Monte Carlo permutation was used to see if the treatment's impact, captured by canonical coefficients, was indeed present at each time point. Four hundred ninety-nine new datasets were generated under the null hypothesis of no treatment effect while maintaining the observed treatment and sampling time structure. Sampling time, replication, and year-replication interactions were treated as covariates in the analysis, restricting data permutation to preserve repeated measures and blocking structure, effectively removing these effects from the residual error. The P-value determined the significance level, calculated as the proportion of F values from the permuted data greater than or equal to the observed F value.

Diversity measurements and abundances of individual or pooled taxa contributing significantly to the overall community response were calculated by the mean values and standard errors. Treatment effects were further evaluated with a mixed-model ANOVA, where seed treatment and sampling time were fixed effects, and replication and year were random effects, repeated measures adjusted for temporally correlated observations. Before analysis, residual plots and the Shapiro-Wilk's W test were applied to assess data normality and homogeneity of variance assumptions. Transformations or variance grouping are used as needed to meet ANOVA assumptions.

## Results

### Effects on pest communities

The effects of insecticidal seed treatments on cotton fluctuated throughout the experiment, showing no consistent pattern (S1 Table). The interaction of the treatment and sampling weeks was changed for the arthropod groups. Across the years, the treatment effects on the combined population of Miridae and Cicadellidae were significant ($p \leq 0.01$). More abundant arthropod groups in the insecticide-treated plot included Cicadellidae and Miridae. The overall mean number of Cicadellidae per 50 sweep net for the first week was recorded as $844 \pm 44.55$ and $995 \pm 12.90$ in the untreated control plots in 2021 and 2022, respectively. Analysis of individual arthropod groups showed a significant reduction in the overall abundance of insecticide-treated plots. In 2021, the abundance of Cicadellidae exhibited marked fluctuations, particularly between the third and eighth weeks of observation during the vegetative and flowering and boll retention stages (S1 Fig). Notably, as Cicadellidae densities decreased, Miridae populations tended to increase, suggesting a possible compensatory dynamic between the two taxa ($p < 0.05$). Clothianidin seed treatment consistently reduced Cicadellidae abundance compared to the untreated control across all phenological stages. However, on six sampling dates 25 June ($F = 5.903$, $p = 0.038$), 1 July ($F = 10.858$, $p = 0.010$), 9 July ($F = 7.869$, $p = 0.021$), 19 July ($F = 68.662$, $p < 0.001$), 5 August ($F = 58.644$, $p = 0.001$), and 19 August ($F = 68.458$; $p < 0.001$) no significant differences were found between the clothianidin and azoxystrobin treatments during the flowering and boll development and opening stages.

In 2022, the Cicadellidae population was consistently more abundant in the untreated control cotton plot, particularly during the vegetative, flowering, and boll retention stages, compared to the CLO seed treatment. Differences between the control and CLO-treated plots were statistically significant throughout the season. However, on 24 June ($F = 2.000$,

p = 0.216), 29 July (F = 2.558, p = 0.157), 12 August (F = 0.139, p = 0.873), and 26 August (F = 2.878, p = 0.133), corresponding to the flowering and boll development and opening stages, no significant differences were detected between the two treatments (S2 Fig).

The mean (±SEM) seasonal densities of Miridae per 50 sweep net for the first week in the untreated control plots were 55 ± 3.53 in 2021 and 93 ± 14.2 in 2022. CLO and AMF suppressed the Miridae population throughout the crop-growing season of 2021. Only on 19 August (F = 71.695, p < 0.001), AMF seed treatment was less effective than before. In the last two weeks of the 2021 cotton growth season, no significant change in population density was observed in the pesticide seed treatment plots compared to the untreated control plots. This may be related to the fact that the last two weeks coincide with the end of the season (S3 Fig).

In 2022, there weren't any essential differences between the population of Miridae in the untreated control group and treated areas for the first (F = 2.160; p = 0.197) and last two weeks (F = 3.000; p = 0.125) and also in the 4 August (F = 1.807; p = 0.243) and 19 August (F = 1.623; p = 0.273) (S1 Table). CLO seed treatment diminished mean seasonal densities for seven weeks of the 2022 growing season. The lower Miridae population observed with the CLO seed treatment compared to the untreated control and AMF seed treatment indicates that CLO played a key role in reducing the pest population (S4 Fig).

Across the 2021 and 2022 growing seasons, the seasonal mean abundances of Miridae and Cicadellidae differed significantly among seed treatments (Table 1). For Miridae, treatment effects were highly significant in 2021 (F = 1871.29, p < 0.001) and remained significant in 2022 (F = 53.52, p = 0.0013). Seasonal comparisons indicated that Miridae abundance was consistently highest in control plots, lowest in clothianidin-treated plots, and moderate in AMF-treated plots across both years.

Similarly, Cicadellidae abundance differed significantly among treatments in both years (F = 276.03, p < 0.001 in 2021; 2022: F = 37.86, p = 0.0025 in 2022). Seasonal patterns were comparable to those observed for Miridae, with control plots showing the highest abundances, clothianidin-treated plots the lowest, and AMF-treated plots moderate values in both growing seasons. Overall, these results demonstrate a consistent treatment effect across years, with clothianidin treatments associated with the greatest reduction in sucking pest populations (Table 1).

## Effects on predator communities

CLO and AMF-seed-treated plots had significantly fewer predators than the untreated control plots. The total predator population was drastically reduced due to seed treatments in 2021. Field treated with clothianidin seed treatment showed a significant reduction in predator population by approximately 43%, from a mean value of 4.39 ± 0.38 predators per sweep net in untreated cotton to 2.48 ± 0.27 in treated fields. Similarly, fields treated with a combination of AMF decreased 26% in the total predator population from a mean value of 4.39 ± 0.38 predators per sweep net in untreated cotton to 3.22 ± 0.33 in treated cotton fields (F = 1.693, p = 0.225). The study again highlighted a considerable effect of seed treatment on the predator population in the following year, 2022. The seed treatment with clothianidin resulted in a 75% reduction in the

**Table 1.** Seasonal mean abundance of Miridae and Cicadellidae under different seed treatments in 2021 and 2022.

| 2021 | Control | CLO | AMF | F | P |
|---|---|---|---|---|---|
| Miridae | 257.67 ± 56.69 a | 107.00 ± 23.71c | 164.00 ± 30.20 b | 1871.29 | <0.001 |
| Cicadellidae | 577.67 ± 69.75 a | 263.67 ± 40.60 c | 409.67 ± 67.56 b | 276.03 | <0.001 |
| **2022** | | | | | |
| Miridae | 231.00 ± 39.37 a | 70.67 ± 14.97 c | 149.00 ± 24.60 b | 53.52 | 37.86 |
| Cicadellidae | 419.00 ± 53.43 a | 192.33 ± 33.82 b | 277.00 ± 40.86 b | 0.0013 | 0.0025 |

Means with row followed by the same letter are not significantly different from each other according to Tukey's HSD test, α = 0.05

predator population from a mean value of 3.38±0.13 per sweep net in untreated cotton to 0.83±0.06 per sweep net in treated fields. Meanwhile, the field treated with the AMF resulted in a 57% decline in the total predator population from a mean value of 3.38±0.13 to 1.43±0.06 predators per sweep net (F=3.933, p=0.049) (Table 2). The taxa that exhibited significant reductions in treated fields across both years included Chrysopidae, Nabidae, Coccinellidae, Anthocoridae, and Asilidae. During the flowering and boll retention stage for both years, the Anthocoridae population peaked at the beginning of August (S5 and S6 Fig). Among the treatments, CLO seed treatment resulted in lower population densities than the untreated control and showed significant differences between the control and AMF seed treatment (Table 2). The total abundance of the Asilidae community was higher in the untreated control group for both years, and in 2021, the CLO seed treatment led to low Asilidae population density compared to AMF seed treatments (F=5.089, p=0.008) (Table 2). During the different development stages, changes in the Asilidae population exhibited inconsistent variation, reaching its highest level in both years in August (S7 and S8 Fig). The total abundance of the Coccinellidae population did not significantly change under pesticide seed treatments in 2021 (F=1.844, p=0.163), but it was lower in 2022 (F=15.386, p=0.001) (Table 2). The abundance of the Coccinellidae population showed inconsistent variation in the development stages for both years. However, the population density peaked in July for both years (S9 and S10 Fig). The abundance of predatory Chrysopidae was reduced in both insecticide treatments compared to the untreated control in 2021 (F=3.149, p=0.001) and in 2022 (F=26.654, p=0.001) (Table 2). It is noteworthy that in both years, seed treatment had a similar variable impact on the abundance of predatory Chrysopidae, while the number of Chrysopidae population reduced in July and peaked between August in both years (S11and S12 Fig). Numbers of Nabidae were significantly lower with CLO and AMF seed treatments in 2021 (F=4.761, p=0.01) and in 2022 (F=15.163, P=0.00) (Table 2). According to the interactions between treatment and plant stage, the Nabidae population was reduced from the vegetative stage through the end of the flowering and boll retention stage in both years (S13and S14 Fig).

## Community-level implications

The principal response curve shows the effects of CLO and AMF-treated cotton seed on the arthropod community for 12 weeks. Analysis was performed with sweep net capture data as plot averages pooled over both study sites. The first ordination axis explained 68.7% of the variance with sweep net data. It indicated that the overall community response

**Table 2. Mean sums of cotton natural enemy population collected per plot using sweep nets.**

| 2021 | Control | CLO | AMF | F | P |
|---|---|---|---|---|---|
| Chrysopidae | 3.50±0.39a | 2.19±0.35b | 2.92±0.35ab | 3.149 | 0.047 |
| Nabidae | 1.83±0.27a | 1.08±0.146b | 1.03±0.17b | 4.761 | 0.010 |
| Coccinellidae | 6.92±1.33a | 3.92±0.84a | 4.92±1.12a | 1.844 | 0.163 |
| Anthocoridae | 6.36±1.02a | 3.44±0.93b | 4.19±0.99ab | 2.396 | 0.096 |
| Asilidae | 3.36±0.42a | 1.78±0.27b | 3.03±0.41a | 5.089 | 0.008 |
| Total Predators | 21.97±0.38a | 12.41±0.27b | 16.09±0.33b | 1.693 | 0.225 |
| 2022 | | | | | |
| Chrysopidae | 3.72±0.81a | 0.86±0.47b | 1.39±0.39b | 26.654 | 0.001 |
| Nabidae | 1.44±0.36a | 0.44±0.28b | 0.53±0.28b | 15.163 | 0.001 |
| Coccinellidae | 1±0.26a | 0.17±0.14b | 0.25±0.16b | 15.386 | 0.001 |
| Anthocoridae | 6.56±0.89a | 1.42±0.36b | 3.25±0.55ab | 2.396 | 0.096 |
| Asilidae | 4.19±0.75a | 1.28±0.38b | 1.75±0.35b | 21.974 | 0.001 |
| Total Predators | 16.91±0.13a | 4.17±0.06b | 7.17±0.06ab | 3.933 | 0.049 |

Means with row followed by the same letter are not significantly different from each other according to Tukey's HSD test, α=0.05

in the insecticide-treated plots significantly deviated from the zero-reference line representing the control community (p≤0.01). Taxon weight indicates which arthropod groups contributed the most to the community response. Higher positive weight indicates that arthropod abundances in the insecticide-treated plots followed the trend depicted by the response curve, whereas higher negative values reveal the opposite. The most affected arthropod groups include Nabidae, Miridae, Anthocoridae, Asilidae, Chrysopidae, Cicadellidae, and Coccinellidae (Fig 1).

Principal response curve (PRC) showing the effects of seed treatments (clothianidin, CLO; azoxystrobin + metalaxyl-M + fludioxonil, AMF) on arthropod community composition over time during the cotton growing seasons. The untreated control is represented by the horizontal baseline (zero line), and treatment effects are expressed as deviations from the control across sampling weeks. Species weights indicate the contribution of each arthropod group to the observed community responses; taxa with higher absolute weights are more strongly associated with treatment effects.

The difference between years in terms of the effect of seed treatments on the Shannon-Wiener and Evenness Indexes of predators were significant. However, seed treatments and seed treatments × year interactions were found to have no significant effect on the Shannon-Wiener and Evenness Index of predators (Table 3). The Shannon-Wiener and Evenness Index were higher in 2021 by about 19.07% and 19.16%, respectively (Table 4).

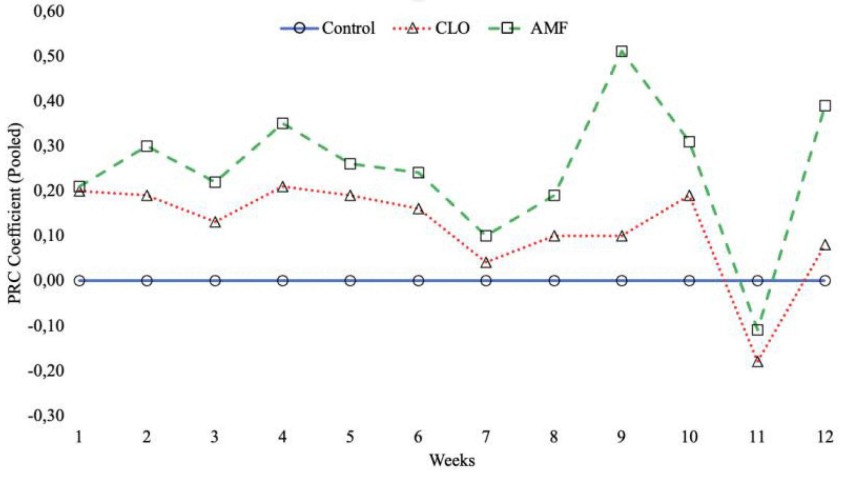

| Species | Pooled |
|---|---|
| Nabidae | -0.33 |
| Miridae | -0.36 |
| Anthocoridae | -0.53 |
| Asilidae | -0.61 |
| Chrysopidae | -0.87 |
| Cicadellidae | -1.07 |
| Coccinellidae | -2.39 |

**Fig 1. Principle response curve (PRC) exhibiting the effects of different seed treatments on arthropod communities during the vegetation period of cotton.**

**Table 3. Results (*F*-values) of repeated-measures (weeks) ANOVA of the effects of seed treatments, years, and their interaction on the Shannon-Wiener and Evenness Index of predators.**

| Community Diversity Indexes | Effects | df | Predators |
|---|---|---|---|
| Shannon-Winer | Year (Y) | 1 | 4.23* |
|  | Seed Treatment (ST) | 2 | 0.54 |
|  | Y × ST | 2 | 0.11 |
| Evenness | Year (Y) | 1 | 4.22* |
|  | Seed Treatment (ST) | 2 | 0.54 |
|  | Y × ST | 2 | 0.11 |

*: $p \leq 0.05$, **: $p \leq 0.01$, no asterisk: nonsignificant differences.

**Table 4. The comparison of seed treatments regarding Shannon-Wiener and Evenness Index in each year separately.**

| Years | Treatments | Shannon-Wiener Index | Evenness Index |
|---|---|---|---|
| 2021 | Control | 1.035 | 0.643 |
| | CLO | 0.938 | 0.583 |
| | AMF | 1.000 | 0.621 |
| | *Year Average* | *0.991 A* | *0.616 A* |
| 2022 | Control | 0.890 | 0.553 |
| | CLO | 0.763 | 0.474 |
| | AMF | 0.751 | 0.467 |
| | *Year Average* | *0.802 B* | *0.498 B* |

Means not followed by the same letter(s) in the same column are significantly different. Upper-case letters were used to compare years with each other.

## Discussion

### Effects of pesticides on pest communities

This experiment investigated the community-level effects of AMF and CLO on arthropods in the cotton agroecosystem. A total of 7 families were recorded in the foliar communities, showing variation among the seed treatments with CLO, AMF, and the untreated control. PRC analyses indicated changes in taxon abundances mainly during the vegetation period of cotton. Overall, in both years (2021 and 2022), the arthropods on cotton plants were more significantly affected by CLO than AMF treatment. Two herbivore taxa were prevalent on cotton in our two-year experiment, including Cicadellidae and Miridae, particularly during the vegetative, flowering, and boll development stages of cotton. These results confirm that soybean-based neonicotinoid seed treatments can significantly suppress early-season herbivore populations [17]. Similarly, imidacloprid-treated cotton has been shown to effectively reduce *Amrasca devastans* populations while causing minimal impact on certain natural enemies [30]. However, AMF treatment had inconsistent effects on pest populations. Especially in 2021, it resulted in moderate suppression of leafhoppers and mirids, though the effect was less evident than that of clothianidin. These findings are consistent with those of [5,31], who observed that fungicide seed treatments are not primarily used for insect control; they may indirectly alter arthropod communities by interacting with environmental conditions and predator dynamics. According to the experiments about pesticide seed treatment effects on arthropod community, herbivores, particularly thrips [25], soybean aphids [32], sugarcane aphids [20], and leafhoppers were negatively affected by clothianidin than other taxon groups, in contrast, some groups such as grasshoppers [17], Lygus sp., Elaterids [18] Collembolans [25] exhibited significantly higher abundances in the seed treated plots. Within the current study, the population dynamics of Cicadellidae highlighted considerable variation between the two years. In 2021, the peak densities of Cicadellidae were primarily observed during the vegetative and boll opening stages, particularly 115–126 days after sowing (DAS), whereas in 2022, the population was more concentrated in vegetative and anthesis stages (initial high densities around 44 DAS, later highest around 86 DAS). The year-to-year variation in pest population showed different environmental conditions in the two growing seasons in Aydın, potentially influencing population dynamics and pest phenology. The comparison of these findings is consistent with the study of Saeed et al. [30], which highlighted similarities in early population development. Saeed et al. [30] reported that *Amrasca devastans* (Hemiptera: Cicadellidae) population peaked at 50 and 55 days post-sowing (DAS) following imidacloprid and thiamethoxam treatment in cotton during 2010 and 2011, respectively, which corresponds closely with the early period of high Cicadellidae density observed around 44–52 DAS in the current study.

The seasonal peak of Miridae abundance, recorded between late July and early August (approximately 84–101 DAS in 2021 and 86–100 DAS in 2022) in the current study, is consistent with findings from other cotton-growing regions [33–35]. Notably, the absence of significant differences in the Miridae population among treatments during the final two weeks of

sampling (beyond ~115 DAS) suggests that the residual efficacy of seed-applied insecticides, particularly CLO, had likely declined by the late boll development stage.

This observation is consistent with the anticipated degradation of systemic insecticides over a growing season and supports findings by Zhang et al. [36], who also reported reduced efficacy of nitenpyram seed treatment against mirid bug *Apolygus lucorum* during later sampling periods.

The variation observed in the current study about the effectiveness of CLO compared to the consistent early-season control could be attributed to differences in the dominant species present in the community, regional environmental factors affecting systemic uptake and persistence, or local agroecological conditions in the region. Therefore, while the population of Cicadellidae and Miridae indicated predictable seasonal peaks, their susceptibility to specific seed treatments like CLO may be variable and suggests further evaluation, explicitly concerning the dominant species and the duration of effective suppression under diverse regional conditions.

## Effects of pesticides on predator communities

The study shows that pesticide seed treatments significantly reduce predatory insect populations, consistent with previous research [37–42]. These predators are essential for regulating pest outbreaks in agroecosystems [14,25,32,43]. Systemic pesticides may disrupt predator communities, weakening natural pest control and increasing pesticide dependence [5,18]. In this study, the neonicotinoid insecticide CLO and the fungicide mixture AMF treatments reduced predator densities over two years, indicating effective translocation within the plant. The movement of systemic pesticides depends on properties such as lipophilicity and log Kow, which enhance their solubility and distribution to various plant parts [11]. This systemic spread increases the risk of exposure to non-target organisms. Furthermore, previous studies emphasize the vulnerability of omnivorous predators such as Geocoris, Orius, and Coccinellids due to their feeding behaviour, such as consuming nectar, pollen, plant tissues, and herbivorous prey; these natural enemies are exposed to both direct and indirect pathways of pesticide uptake. For example, indirect poisoning can occur when predators consume herbivores like leafhoppers that have ingested neonicotinoids [10,30,43,44]. Additionally *Lysiphlebus testaceipes* (Cresson) (Hymenoptera: Braconidae) populations were significantly reduced after exposure to thiamethoxam contaminated extrafloral nectar [16].

Particularly during the early vegetative stage of the plant, the higher mortality of natural enemies can be explained by two factors: the intense residue levels due to the plant's vegetative stage and the fact that predators are in their immature stages, during which they have a greater need for plant nutrients [11,45,46]. According to the experiment of Gontijo et al. [11], the insecticidal toxicity of thiamethoxam and clothianidin was diminished with plant development, and thiamethoxam-coated seed caused lethal and sublethal effects on *O. insidiosus* (Hemiptera: Anthocoridae) while clothianidin-coated seed led to sublethal impacts. In a study conducted on imidacloprid and thiamethoxam treatment on soybean seed, the general predator community was reduced by 25%, and similar predators, with the present study, such as *O. insidiosus*, *Nabis americoferus* (Hemiptera: Nabidae), and *Chrysoperla* (Neuroptera: Chrysopidae) populations were reduced in the pesticide-treated plots [17]. The current study results coincide with previous findings and demonstrate even more significant reductions in predator abundance, 44% in 2021 and 75% in 2022. These reductions surpass the 34.5% decrease observed in foliage communities within clothianidin-treated corn agroecosystems [25] and the 16% reduction reported in a meta-analysis by Douglas and Tooker [5].

## Effects of pesticides on arthropod communities

Pesticides reduced the pest population in cotton crops in the experiment, and the lack of pest population in sunflower, corn, and soybean was detected in the other studies [14, 17,18,32]. Pests' responses to pesticide treatment differ according to insect abundance and diversity [18,47].

Using PRC analysis, the current study evaluated the community-level effects of CLO and AMF seed treatments on arthropods in an Aydın cotton agroecosystem. This study provides novel field-based evidence that seed-applied systemic

pesticides can influence not only target sucking pest populations but also the abundance and diversity of associated predator communities in cotton agroecosystems. By simultaneously examining pest and predator assemblages, the present work contributes to a more comprehensive understanding of how prophylactic seed treatments shape multitrophic arthropod interactions under field conditions. According to the PRC results, certain predatory insect families—most notably Coccinellidae, Chrysopidae, Asilidae, Anthocoridae, and Nabidae—as well as pest groups such as Cicadellidae and Miridae, exhibited significant negative responses to the pesticide in terms of overall predator species richness. However, the difference in predator diversity between years suggests that factors such as seed treatments, cropping practices, prey availability, or other abiotic influences such as climate variability may have negatively impacted predator communities [48]. Additionally, the evenness index in the present study declined from 0.616 in 2021 to 0.498 in 2022, indicating a reduction in the balance of species distribution over time. This finding contrasts with the higher evenness values reported by Firake and Behere [27] and Disque et al. [25], aligning with prior research suggesting that seed-applied insecticides can adversely impact the predator community. Consistent with previous studies, clothianidin treatments affected predator abundance more than overall diversity [5,25,44]. Predator diversity and evenness significantly decreased from 2021 to 2022, with untreated control plots consistently exhibiting the highest diversity and evenness values, showing the disruptive effect of seed-applied pesticides on predator communities. This study's Shannon-Wiener diversity index values ranged from 0.751 to 1.035, with annual means of 0.991 in 2021 and 0.802 in 2022. These values were considerably lower than those reported by Firake and Behere [27] in northeast India, where Shannon index values typically exceeded 2.16, reflecting higher predator diversity in maize agroecosystems. Overall, these findings suggest that while seed-applied pesticides may not drastically reduce predator species richness at the community level, they can suppress predator abundance and disrupt the evenness of arthropod predator communities. From an agroecosystem sustainability perspective, these results highlight the importance of moving beyond target pest suppression when evaluating the ecological consequences of seed-applied systemic pesticides in cotton production systems. These findings provide an ecological basis for improving integrated pest management strategies by emphasizing the importance of balancing effective pest suppression with the conservation of beneficial predator communities in cotton agroecosystems.

## Conclusion

Pesticide seed treatments are commonly applied in fields without consistent pest pressure as a precaution against pest outbreaks. CLO and AMF seed treatments significantly reduced arthropod populations in the Aydın cotton agroecosystem, with communities recovering over the growing season. The reduction in the Cicadellidae population leads to an increased Miridae population. Despite impacts on pest abundance, the pesticide seed treatment reduced the natural enemy population. The Shannon-Wiener diversity index and evenness of predators were analyzed for two consecutive years. The results showed no significant effect of CLO and AMF seed treatment on community diversity and evenness, but diversity and evenness were substantial over the years. The difference between years may be cropping practices, prey availability, or other abiotic influences, such as climate variability, that may have negatively impacted predator communities. According to the principal response curve analyses, the most affected arthropod group from seed treatment was observed: Nabidae, Miridae, Anthocoridae, Asilidae, Chrysopidae, Cicadellidae, and Coccinellidae. These findings highlight the importance of continuously and systematically monitoring pest and predator populations at the community level in cotton agroecosystems. Future research integrating species-level identification, longer-term field monitoring, and assessments of sublethal or behavioural effects on natural enemies would further improve understanding of the ecological consequences of prophylactic seed treatments and their implications for multitrophic interactions and biological control within sustainable IPM-based cotton production systems.

## Supporting information

**S1 Table. Mean sums of Cicadellidae and Miridae populations collected per plot using sweep nets.**
(PDF)

**S2 Table. Mean (± SD) abundance of arthropod families (Miridae, Cicadellidae, Anthocoridae, Asilidae, Coccinelli-dae, and Nabidae) under different seed treatments across sampling weeks in 2021 and 2022.**
(PDF)

**S1 Fig. Impact of pesticide treatments on Cicadellidae abundance during crop development stages in 2021.**
(PDF)

**S2 Fig. Impact of pesticide treatments on Cicadellidae abundance during crop development stages in 2022.**
(PDF)

**S3 Fig. Impact of pesticide treatments on Miridae abundance during crop development stages in 2021.**
(PDF)

**S4 Fig. Impact of pesticide treatments on Miridae abundance during crop development stages in 2022.**
(PDF)

**S5 Fig. Impact of pesticide treatments on Anthocoridae abundance during crop development stages in 2021.**
(PDF)

**S6 Fig. Impact of pesticide treatments on Anthocoridae abundance during crop development stages in 2022.**
(PDF)

**S7 Fig. Impact of pesticide treatments on Asillidae abundance during crop development stages in 2021.**
(PDF)

**S8 Fig. Impact of pesticide treatments on Asillidae abundance during crop development stages in 2022.**
(PDF)

**S9 Fig. Impact of pesticide treatments on Coccinellidae abundance during crop development stages in 2021.**
(PDF)

**S10 Fig. Impact of pesticide treatments on Coccinellidae abundance during crop development stages in 2022.**
(PDF)

**S11 Fig. Impact of pesticide treatments on Chrysopidae abundance during crop development stages in 2021.**
(PDF)

**S12 Fig. Impact of pesticide treatments on Chrysopidae abundance during crop development stages in 2022.**
(PDF)

**S13 Fig. Impact of pesticide treatments on Nabidae abundance during crop development stages in 2021.**
(PDF)

**S14 Fig. Impact of pesticide treatments on Nabidae abundance during crop development stages in 2022.**
(PDF)

## Author contributions

**Conceptualization:** Melis Yalçın.

**Data curation:** Melis Yalçın.

**Formal analysis:** Melis Yalçın.

**Funding acquisition:** Melis Yalçın.

**Investigation:** Melis Yalçın.

**Methodology:** Melis Yalçın.

**Project administration:** Melis Yalçın.

**Resources:** Melis Yalçın.

**Software:** Melis Yalçın.

**Supervision:** Melis Yalçın.

**Validation:** Melis Yalçın.

**Visualization:** Melis Yalçın.

**Writing – original draft:** Melis Yalçın.

**Writing – review & editing:** Melis Yalçın.

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
