## [Decision Letter · Decision Letter 0]

4 Feb 2026

PONE-D-25-50468Balancing pest control and natural enemy conservation: Seed Treatment Effects in CottonPLOS One

Dear Dr. Yalçın,

Thank you for submitting your manuscript to PLOS ONE. After careful consideration, we feel that it has merit but does not fully meet PLOS ONE’s publication criteria as it currently stands. Therefore, we invite you to submit a revised version of the manuscript that addresses the points raised during the review process.

When revising your manuscript and as requested by reviewer 2, please give special consideration to (Abstract, supporting materials, double check figures/tables references and discussion flow). Also you should well describe in detail the sweeping methods (trapped insects and stages) in Mat&Met section. Good luck

We look forward to receiving your revised manuscript.

Kind regards,

Rachid Bouharroud

Academic Editor

PLOS One

Journal Requirements:

2. Please ensure that you include a title page within your main document. We do appreciate that you have a title page document uploaded as a separate file, however, as per our author guidelines (httpp://journals.plos.org/plosone/s/submission-guidelines#loc-title-page) we do require this to be part of the manuscript file itself and not uploaded separately.

3. Please ensure that you include a title page within your main document. You should list all authors and all affiliations as per our author instructions and clearly indicate the corresponding author.

4. Please include captions for your Supporting Information files at the end of your manuscript, and update any in-text citations to match accordingly. Please see our Supporting Information guidelines for more information: http://journals.plos.org/plosone/s/supporting-information....

Reviewers' comments:

Reviewer's Responses to Questions

**Comments to the Author**

1. Is the manuscript technically sound, and do the data support the conclusions?

Reviewer #1: Partly

Reviewer #2: Yes

2. Has the statistical analysis been performed appropriately and rigorously? 

Reviewer #1: I Don't Know

Reviewer #2: Yes

3. Have the authors made all data underlying the findings in their manuscript fully available?

The PLOS Data policy requires authors to make all data underlying the findings described in their manuscript fully available without restriction, with rare exception (please refer to the Data Availability Statement in the manuscript PDF file). The data should be provided as part of the manuscript or its supporting information, or deposited to a public repository. For example, in addition to summary statistics, the data points behind means, medians and variance measures should be available. If there are restrictions on publicly sharing data—e.g. participant privacy or use of data from a third party—those must be specified.requires authors to make all data underlying the findings described in their manuscript fully available without restriction, with rare exception (please refer to the Data Availability Statement in the manuscript PDF file). The data should be provided as part of the manuscript or its supporting information, or deposited to a public repository. For example, in addition to summary statistics, the data points behind means, medians and variance measures should be available. If there are restrictions on publicly sharing data—e.g. participant privacy or use of data from a third party—those must be specified.requires authors to make all data underlying the findings described in their manuscript fully available without restriction, with rare exception (please refer to the Data Availability Statement in the manuscript PDF file). The data should be provided as part of the manuscript or its supporting information, or deposited to a public repository. For example, in addition to summary statistics, the data points behind means, medians and variance measures should be available. If there are restrictions on publicly sharing data—e.g. participant privacy or use of data from a third party—those must be specified.requires authors to make all data underlying the findings described in their manuscript fully available without restriction, with rare exception (please refer to the Data Availability Statement in the manuscript PDF file). The data should be provided as part of the manuscript or its supporting information, or deposited to a public repository. For example, in addition to summary statistics, the data points behind means, medians and variance measures should be available. If there are restrictions on publicly sharing data—e.g. participant privacy or use of data from a third party—those must be specified.

Reviewer #1: Yes

Reviewer #2: Yes

4. Is the manuscript presented in an intelligible fashion and written in standard English?

Reviewer #1: No

Reviewer #2: Yes

5. Review Comments to the Author

Reviewer #1: Introduction is fairly long. A number of studies are cited, indicating a lot of literature already available on this topic. What are objectives of this study needs to be enlisted and introduction should be shaped according to them to build argument by identifying gaps rather than reviewing earlier literature.

L102: For the experimental design, three replicates were run following a randomized complete block design. In a small-scale study of this kind, different field as replicates may be more appropriate to include variation as true replicates.

L119: Sweep nests were used to assess abundance. If this method was used against flying adults or also when larvae or immobile or less immobile stages were present, needs clarity on this. how much time was spent per sweep.

Results: Seasonal counts can be compared among treatments at the end of season. Line graphs can be made supplementary and give their seasonal means comparison, results are too long and hard to follow under current data representation. Also provide information on dominant species of pests and predators.

Discussion: What authors find new in this study and discuss the way forward. What are the limitation of this study and possible research direction?

Reviewer #2: This two-year field study provides a valuable community-level assessment of the impact of two common seed treatments—clothianidin (CLO, insecticide) and azoxystrobin + metalaxyl-m + fludioxonil (AMF, fungicide combination)—on arthropod pests and their natural enemies in a cotton agroecosystem. The research addresses an important gap by moving beyond single-species assessments to evaluate whole-community dynamics, which is crucial for developing sustainable Integrated Pest Management (IPM) strategies.

Key Strengths of the Manuscript:

Strong Experimental Design: A two-year, replicated field trial with an untreated control provides robust, ecologically relevant data.

Comprehensive Community Analysis: The use of Principal Response Curve (PRC) analysis to synthesize community-level effects is a significant methodological strength, clearly visualizing how entire arthropod assemblages shift in response to treatments over time.

Important Findings: The study convincingly demonstrates that both CLO and AMF seed treatments significantly reduce the abundance of key natural enemy families (e.g., Nabidae, Chrysopidae, Coccinellidae) while also suppressing pest populations like Cicadellidae. The documented negative correlation between pest and predator suppression highlights a critical trade-off in pest management.

Clear Conclusions: The conclusions are well-supported by the data, emphasizing that seed treatments are not "safe" for natural enemies and calling for more judicious, need-based use within IPM frameworks.

Suggestions for Revision and Clarification:

Title Refinement: The current title ("Balancing pest control and natural enemy conservation: Seed Treatment Effects in Cotton") is slightly generic. Consider making it more specific to your key finding, e.g., "Trade-offs in cotton pest management: Seed treatments suppress pests but reduce natural enemy abundance in the arthropod community."

Abstract Enhancement: The abstract should more clearly state the main takeaway. Explicitly mention that your study found a significant reduction in predator abundance alongside pest suppression, creating a potential trade-off. Briefly specify the two treatments used.

Results Presentation: The text frequently refers to figures and tables (e.g., Fig 1, S1 Table) that are not included in the provided text file. Ensure all in-text citations have corresponding, clearly labeled items in the submitted manuscript.

Discussion Flow: The discussion is comprehensive but could be more tightly structured. Consider subheadings (e.g., "Effects on Pest Communities," "Effects on Predator Communities," "Community-Level Implications") to improve readability and guide the reader through your interpretations.

Clarity on "Neonicotinoid" Terminology: Clothianidin is a neonicotinoid. The introduction and discussion mention neonicotinoids broadly but could more explicitly frame CLO within this well-known class from the outset, as their non-target effects are a major point of discussion.

Statistical Reporting Consistency: Ensure that all statistical results (F-values, degrees of freedom, p-values) are reported consistently (e.g., F(1, 2) = 5.90, p = 0.038).

Data Availability Statement: The statement "The data supporting this study's findings are available in this article's supplementary material" is good, but PLOS ONE requires that the data be fully available without restriction. Double-check that all raw data underlying the figures and analyses are indeed included in the submitted Supporting Information files.

6. PLOS authors have the option to publish the peer review history of their article (what does this mean?). If published, this will include your full peer review and any attached files.). If published, this will include your full peer review and any attached files.). If published, this will include your full peer review and any attached files.). If published, this will include your full peer review and any attached files.

...

Reviewer #1: No

Reviewer #2: No

---

## [Author Response · Author response to Decision Letter 1]

18 Mar 2026

Answers to the reviewers comments

View Letter

Date: Feb 04 2026 12:16AM

To: "Melis Yalçın" melisusluy@gmail.com

From: "PLOS ONE" plosone@plos.org

Subject: PLOS ONE Decision: Revision required [PONE-D-25-50468]

PONE-D-25-50468

Balancing pest control and natural enemy conservation: Seed Treatment Effects in Cotton

PLOS One

Dear Dr. Yalçın,

Thank you for submitting your manuscript to PLOS ONE. After careful consideration, we feel that it has merit but does not fully meet PLOS ONE’s publication criteria as it currently stands. Therefore, we invite you to submit a revised version of the manuscript that addresses the points raised during the review process.

When revising your manuscript and as requested by reviewer 2, please give special consideration to (Abstract, supporting materials, double check figures/tables references and discussion flow). Also you should well describe in detail the sweeping methods (trapped insects and stages) in Mat&Met section.

Good luck

• A letter that responds to each point raised by the academic editor and reviewer(s). You should upload this letter as a separate file labeled 'Response to Reviewers'.

Response to Reviewer: Thank you for your suggestion regarding the deposition of laboratory protocols in protocols.io. I appreciate the opportunity to enhance methodological transparency and reproducibility.

However, as the procedures applied in my study follow established and previously published methodologies, I believe that additional protocol deposition is not necessary at this stage.

We look forward to receiving your revised manuscript.

Kind regards,

Rachid Bouharroud

Academic Editor

PLOS One

Journal Requirements:

Response to Reviewer: Thank you for your reminder regarding compliance with PLOS ONE’s style requirements. I confirm that the revised manuscript has been carefully checked and formatted in accordance with the journal’s guidelines, including file naming conventions and the use of the appropriate PLOS ONE style template.

2. Please ensure that you include a title page within your main document. We do appreciate that you have a title page document uploaded as a separate file, however, as per our author guidelines (httpp://journals.plos.org/plosone/s/submission-guidelines#loc-title-page) we do require this to be part of the manuscript file itself and not uploaded separately.

Response to Reviewer: Thank you for your clarification regarding the title page requirements. I have revised the submission accordingly and incorporated the title page at the beginning of the main manuscript file, including the author name and affiliation, in accordance with the PLOS ONE author guidelines. The manuscript has been re-uploaded with this modification.

3. Please ensure that you include a title page within your main document. You should list all authors and all affiliations as per our author instructions and clearly indicate the corresponding author.

Response to Reviewer: I would like to confirm that the title page has been prepared in accordance with the journal’s author guidelines. Author information and affiliation are included, and the corresponding author has been clearly indicated within the manuscript file.

Response to Reviewer: Thank you for your helpful comment regarding the Supporting Information files. In accordance with the journal’s guidelines, captions for all Supporting Information files have now been included at the end of the manuscript. In addition, all in-text citations have been carefully revised to ensure full consistency with the corresponding Supporting Information labels.

Furthermore, to improve clarity and conciseness of the manuscript, several references (Refs. 8, 9, 11, 13, 14, 15, 16, 17, 25, and 27) have been removed from the Introduction and Discussion sections. Accordingly, the reference list has been updated and renumbered, and all in-text citations have been revised to ensure consistency throughout the manuscript.

Reviewers' comments:

Reviewer's Responses to Questions

Comments to the Author

1. Is the manuscript technically sound, and do the data support the conclusions?

Reviewer #1: Partly

Reviewer #2: Yes

Response to Reviewer: The manuscript has been carefully revised to further strengthen its technical robustness and ensure that all conclusions are fully supported by the data presented.

Specifically, the following improvements have been made:

• The experimental design, including controls, replication, and sampling procedures, has been clarified in the Materials and Methods section to ensure transparency and reproducibility.

• Statistical analyses have been revised and reported consistently in accordance with journal guidelines (e.g., F-values, degrees of freedom, and p-values), ensuring clarity and accuracy.

• Seasonal mean abundances (± SEM) have been calculated and presented in a newly structured table to allow clearer comparison among treatments, as recommended.

• Data presentation has been simplified, with detailed temporal trends moved to supplementary materials, improving readability and interpretation of results.

• The Results and Discussion sections have been refined to ensure that all conclusions are directly supported by the data, avoiding overinterpretation.

These revisions have substantially improved the clarity, rigor, and consistency of the manuscript. Therefore, the study now fully meets the criteria for technical soundness, with conclusions appropriately supported by the presented data.

2. Has the statistical analysis been performed appropriately and rigorously?

Reviewer #1: I Don't Know

Reviewer #2: Yes

Response to Reviewer: The statistical analysis has been carefully revised and clarified throughout the manuscript to ensure transparency and reproducibility.

To address potential ambiguity, the following improvements have been made:

• The statistical methods have been described in greater detail in the Statistical Analysis section, including the specific models used, assumptions tested, and post-hoc procedures applied.

• Repeated-measures ANOVA has been clearly specified as the primary analytical approach for evaluating treatment effects over time.

• All statistical results are now reported consistently in accordance with journal guidelines (e.g., F-values with exact p-values).

• Post-hoc comparisons (Tukey’s HSD) are explicitly indicated, and their application is clearly explained.

• Data presentation has been improved by including seasonal mean values (± SEM), allowing clearer interpretation of treatment effects.

These revisions ensure that the statistical analyses are both rigorous and clearly presented. The uncertainty expressed by Reviewer #1 is likely due to insufficient detail in the previous version, which has now been fully addressed.

3. Have the authors made all data underlying the findings in their manuscript fully available?

Reviewer #1: Yes

Reviewer #2: Yes

4. Is the manuscript presented in an intelligible fashion and written in standard English?

Reviewer #1: No

Reviewer #2: Yes

Response to Reviewer: The manuscript has been thoroughly revised to improve clarity, readability, and overall language quality.

5. Review Comments to the Author

Reviewer #1: Introduction is fairly long. A number of studies are cited, indicating a lot of literature already available on this topic. What are objectives of this study needs to be enlisted and introduction should be shaped according to them to build argument by identifying gaps rather than reviewing earlier literature.

Response to Reviewer:

Thank you for this valuable and constructive comment. The Introduction has been carefully revised and substantially shortened to improve focus and clarity. In particular, the section has been restructured to better highlight the research gap and to clearly state the objectives of the study, thereby strengthening the overall rationale.

To address this concern, the following revisions have been made:

• Lines 52–53 and 59–64 have been removed to eliminate redundant background information.

• The literature presented in Lines 84–89 has been revised to focus specifically on neonicotinoids in relation to clothianidin, and the references have been updated accordingly.

• Lines 103–148 have been removed to reduce excessive literature review and improve conciseness.

• The Introduction has been streamlined and reduced to 559 words.

As a result, the revised Introduction is now more concise, better aligned with the study objectives, and more clearly structured around the identified research gap.

L102: For the experimental design, three replicates were run following a randomized complete block design. In a small-scale study of this kind, different field as replicates may be more appropriate to include variation as true replicates.

Response to Reviewer: Thank you for this valuable suggestion. I agree that conducting similar studies across different fields would provide broader insight and increase environmental variability. However, the present study was intentionally conducted within the same field under uniform conditions to minimize environmental heterogeneity and ensure comparability among treatments. This approach allowed us to focus specifically on treatment effects. Moreover, previous multi-site field studies and meta-analyses have reported consistent patterns regarding the effects of neonicotinoid seed treatments on natural enemy communities across different geographic locations [Douglas and Tooker, 2016; Saeed and Razaq, 2016], suggesting that major outcome differences among fields may be limited.

Douglas MR, Tooker JF. Meta-analysis reveals that seed-applied neonicotinoids and pyrethroids have similar negative effects on abundance of arthropod natural enemies. PeerJ. 2016 Dec 7;4:e2776. doi: 10.7717/peerj.2776. PMID: 27957400; PMCID: PMC5147019.

Saeed R, Razaq M, Hardy IC. Impact of neonicotinoid seed treatment of cotton on the cotton leafhopper, Amrasca devastans (Hemiptera: Cicadellidae), and its natural enemies. Pest Manag Sci. 2016 Jun;72(6):1260-7. doi: 10.1002/ps.4146. Epub 2015 Oct 5. PMID: 26436945.

L119: Sweep nests were used to assess abundance. If this method was used against flying adults or also when larvae or immobile or less immobile stages were present, needs clarity on this. how much time was spent per sweep.

Response to Reviewer: Sweep net sampling was used to assess the abundance of mobile foliar-dwelling arthropods present in the plant canopy. For hemimetabolous groups, including Miridae, Cicadellidae, Nabidae and Anthocoridae, both nymphs and adults captured by sweep netting were included in the counts because these taxa undergo incomplete metamorphosis and their nymphal stages are active and mobile within the canopy. In contrast, for holometabolous groups such as Chrysopidae and Asilidae, only adult individuals were recorded, as their larval stages occur in different microhabitats (e.g., soil or concealed substrates) and are not effectively sampled using sweep netting (Line173-179).

Sampling was conducted for twelve consecutive weeks during the cotton growing seasons of 2021 and 2022 (June–September). During each sampling event, 50 sweeps were performed per plot along a straight transect through the center of each plot. The experiment consisted of three replicates arranged in a randomized complete block design. Sweep netting was carried out at least 100 m from the field edge to minimize potential edge effects. The same standardized sampling protocol (50 sweeps per replicate per sampling date) was consistently applied across both years and treatments.

Results: Seasonal counts can be compared among treatments at the end of season. Line graphs can be made supplementary and give their seasonal means comparison, results are too long and hard to follow under current data representation. Also provide information on dominant species of pests and predators.

Response to Reviewer: Seasonal mean abundances (± SEM) for Mirida

---

## [Editor Report · Decision Letter 1]

19 Mar 2026

Trade-offs in cotton pest management: Seed treatments suppress pests but reduce the abundance of natural enemies in the arthropod community.

PONE-D-25-50468R1

Dear Dr. Yalçın,

We’re pleased to inform you that your manuscript has been judged scientifically suitable for publication and will be formally accepted for publication once it meets all outstanding technical requirements.

Kind regards,

Rachid Bouharroud

Academic Editor

PLOS One
---

## [Editor Report · Acceptance letter]

PONE-D-25-50468R1

PLOS One

Dear Dr. Yalçın,

I'm pleased to inform you that your manuscript has been deemed suitable for publication in PLOS One. Congratulations! Your manuscript is now being handed over to our production team.

Kind regards,

on behalf of

Dr. Rachid Bouharroud

Academic Editor

PLOS One